# Gender differences in dictator giving: A high-power laboratory test

**Iván Barreda-Tarrazona**, **Ainhoa Jaramillo-Gutiérrez**, **Marina Pavan***,
**Gerardo Sabater-Grande**

Laboratory for Experimental Economics & Economics Department, Jaume I University, Castellón de la Plana, Castellón, Spain

\* pavan@uji.es

## Abstract

We gather information from a large laboratory sample comprising 1,161 subjects and study gender differences in altruism using a dual-role dictator game. We control for factors potentially affecting the role of gender in dictator giving, such as the subject's age, cognitive ability, and personality traits, together with the dictator's self-reported emotions motivating the decision, and response time. We find that women behave in a significantly more generous way than men: after controlling for the factors mentioned above, females transfer 7.5 percentage points (about 40%) more of their endowment than males, on average. Moreover, we find that gender differences in giving are mediated by reasoning ability, personality traits and emotions.

## Introduction

Do females behave more altruistically than males? The answer to this question is important to understand behavior wherever decisions might be influenced by the subject's altruism, from charitable giving to intergenerational transfers, from bargaining to household decision making [1]. A vast literature explored gender differences in altruism in the past, with mixed results [2]. In their recent meta-analysis, Bilén et al. [3] underline that many of these previous studies are underpowered and highlight the need for substantially larger sample sizes. To address this issue, we gather and analyze a large dataset on female and male transfers in the Dictator Game (DG), together with a rich set of other potentially moderating variables, obtained in the controlled environment of an experimental economics laboratory.

Altruistic behavior is the sacrifice of one's resources for the benefit of another individual [4]. Focusing on the concept of altruism as expressed by the willingness to give money, the dictator game is a behavioral paradigm commonly used as an effective method to measure altruism [5]. The dictator game is positively correlated to altruistic acts in real-life situations (e.g., returning money to others, in [6]), charitable giving [7] and willingness to help in a real-effort task [8]. Additionally, Carpenter et al. [9] find that DG giving is positively correlated with the specific survey measures of altruism used in their study. Empirical evidence has shown that dictators give on average 28% of the endowment [3, 10]. However, there is considerable heterogeneity among DG offers, which could potentially be explained by individual characteristics. In this vein, our objective is to study gender differences in dictator's giving controlling for

**Data Availability Statement:** The dataset generated for this study can be found at the Open Science Framework repository: https://osf.io/

g6wej/?view_only=
f3e684ca2fc3424da02215c18088ecac.

**Funding:** This work was supported by the Spanish Ministry of Science and Innovation (PID2021-123053OB-I00) and Jaume I University (UJI-B2021-23). The funders had no role in study design, data collection and analysis, decision to publish, or preparation of the manuscript.".

**Competing interests:** The authors have declared that no competing interests exist.

age, reasoning ability and personality traits, together with the dictator's self-reported motivation for the decision and response time.

Our data were collected as part of laboratory experiments on trust and cooperation conducted in the last four years. All these experiments had a common first phase which elicited dictator giving, cognitive ability, personality traits and other individual characteristics. Using the methodology of experimental economics to conduct well incentivized and properly controlled laboratory experiments greatly facilitates the identification of causal relationships in comparison to other approaches (as surveys, online or field experiments).

Considering the gender effect size reported in Bilén et al. [3], we planned to reach a power of 80% in a t-test with a significance level of 0.05. As underlined above, previous empirical evidence on gender differences in the Dictator Game has not been conclusive on whether males or females are more generous. For instance, Bolton and Katok [11], Ben-Ner et al. [12] and Chowdhury et al. [13] found no significant gender differences in dictator's giving when the gender of the receiver was unknown. On the contrary, in an early study, Eckel and Grossman [14] found that women were more generous than men in a double-blind Dictator Game. Andreoni and Vesterlund [1] arrived at the same conclusion eliminating the double-blind design feature, but only when giving was expensive (i.e. for a price of giving equal to 1 or more). The meta-analyses by Engel [10], Bilén et al. [3], and Doñate-Buendía et al. [15] all suggest that women generally share more than men, while, according to Niederle [2], existing data from dictator games are inconclusive about whether males or females are more generous, implying that we should be cautious about positive results bias. Niederle [2] also advocates for the investigation of the robustness of experimental results. Whenever previous studies on gender differences in Dictator Games involved large samples, it was at the cost of reduced control, or reduced incentives, or both. Brañas-Garza [16] performed a large lab-in-the-field experiment with 2,500 adolescents in Spain playing three binary modifications of the dictator game and found that girls chose more egalitarian distributions of the endowment than boys. However, the payoffs for that study had to be hypothetical in order to obtain permission from the high schools involved. Horn et al. [17] performed another relatively large study with 1,088 adolescents in Hungary, which included dictator giving, and found girls to give significantly more. In their case, the payoffs were in the form of meal vouchers. Large sample studies have also been conducted online: for instance, Brañas et al. [18] found that women give more than men in a dictator game performed by 3,583 United States residents recruited through Amazon M-Turk. The dictators in their study were endowed with just 20 cents. The Global Preference Survey presented in Falk et al. [19] and collecting data from 80,000 participants in 76 countries also contained two altruism related items: a hypothetical charity donation decision and a self-assessment of the willingness to give to good causes. They found that altruism was more pronounced among women. The survey questions were chosen based on their explanatory power of the decisions of 409 experimental subjects participating in three binary Charity Dictator Games, but the incentives at stake amounted to only 2.4 euros. We contribute to this debate by providing the largest number of independent observations (1,161) in a saliently incentivized dictator game economic experiment to date with the increased control of brick-and-mortar laboratory conditions.

The next section of the paper presents and motivates our experimental design and procedures. Subsequently, we describe our results, and finally we conclude.

## Experimental design and methods

The dataset generated for this study can be found at the Open Science Framework repository: https://osf.io/g6wej/?view_only=f3e684ca2fc3424da02215c18088ecac. The experimental

protocol followed was approved by the Deontological Commission of Universitat Jaume I (CEISH/17/2022). Participants gave their written informed consent by registering in the ORSEE [20] database of the Laboratory for Experimental Economics of our university after having read the rules of the experimental sessions in our laboratory and the data protection policy, available at https://www.leerecs.uji.es/orsee/public/privacy.php. They also had the possibility of accepting or not the invitation to the specific sessions. No minors were participating in this study.

Our dataset originated as the first phase of different experiments on cooperation and trust performed at the Laboratory for Experimental Economics (LEE) of Jaume I University in the last four years. The first phase of these experiments was always identical and elicited individual characteristics of the subjects such as gender and age, together with measures of altruism (DG giving), reasoning ability, emotions and personality. It was only in a second phase, about one week later, that the experiments on cooperation/trust were run.

In particular, the experiments were run in 2020, 2022 and 2023, always following the exact same experimental protocol in all 32 first-phase sessions: 6 sessions from November 16th to 18th, 2020; 6 sessions from April 1st to 5th, 2022; 10 sessions from May 15th to June 10th, 2023; and 10 sessions from September 19th to 28th, 2023. The respective recruitments were always done a week before through ORSEE [20], verifying that the subjects had not participated in previous sessions. Subjects at the LEE are university students from different degrees, including undergraduate, graduate, and adult programs.

Inspired by the observation in Bilén et al. [3] that most studies on gender differences in dictator giving were underpowered, we realized that we could build a moderately large dataset with our existing observations on dictator giving in phase 1 of our experiments. These data would allow us to conduct a high-power test of the hypothesis that females give more than males in the Dictator Game. Bilén et al. [3] found a four-percentage points difference (between a giving of 30% of the endowment for males and 34% for females), with a 0.25 standard deviation (Cohen's d of 0.16). Assuming that gender difference effects would be as large as in Bilén et al. [3], in order to get an ex-ante power of 80% and a confidence level of 95% for a one-tailed t-test, a collection of 484 male and 484 female observations was planned. Used measures, procedure and main analysis were pre-registered in July 2023 as "Gender differences in dictator giving" (AsPredicted #138991, https://aspredicted.org/CC6_CK8), in accordance with current methodological recommendations [21]. In July 2023 we had only 318 male and 431 female observations, hence we continued gathering data until we reached the threshold of 484 for both sexes by October 2023. Finally, we obtained independent observations on a total of 1,161 (675 female and 486 male) participants. Sessions were approximately gender-balanced, with about 60 participants each. Subjects played both in the role of dictator (which we more neutrally called "sender") and that of recipient. As dictators, they were asked how much of their endowment of 10€ to transfer to an anonymous recipient they were randomly paired with, knowing that the rest of the endowment would be for them. As recipients, they were randomly matched with another anonymous subject, and did not have to do anything. Subjects did not know whether they were paired with a male or a female in any of the two roles that they were playing. Moreover, they were told that the dictator they were paired with would be a different person from their recipient, which minimized reciprocity concerns. At the end of the experiment, they were paid for one of the two roles, randomly chosen. According to Zahn et al. [22], the two possible differential effects of using a dual role instead of a single-role version of the dictator game are an increased empathy towards the counterpart and a decreased responsibility regarding the final payoffs. Our goal in this study is not finding out which of these two likely effects prevails in our sample, but just testing for gender differences in dictator giving keeping the game implementation invariant.

At the beginning of each session, subjects were given written instructions (a translation of these instructions into English can be found in the S1 File), which were also read aloud by the organizers. Any remaining questions were privately answered. Average earnings were around 11.5€ and each session lasted about one hour. Experiments were computerized and carried out in a specialized computer lab (LEE at Jaume I University), using software based on the Z-Tree toolbox by Fischbacher [23].

## Assessing reasoning ability, personality traits and emotions

Based on the extensive previous literature, apart from gender, we also elicited a series of other factors that could potentially explain individual differences in dictator giving: age, previous experience with laboratory experiments, measures of reasoning ability and personality traits, and the self-reported nature of the motivation for the decision, distinguishing between rationality and emotions. We also measured each subject's response time.

The DG is a game involving no strategic reasoning. In fact, Brandstätter and Güth [24] argue that the dictator's behavior is more a motivational and emotional problem than an intellectual one. In that sense, from a neoclassical point of view, giving high positive amounts could be interpreted as errors by misunderstanding of instructions. These errors are less likely to be made by subjects with stronger cognitive abilities. Following this argument, there could be a significant negative relationship between the dictator's giving and cognitive ability. Another possibility is that cognitive ability is systematically related to economic preferences and hence to generosity [19].

We tested for reasoning ability using the Abstract Reasoning part of the Differential Aptitude Test for Personnel and Career Assessment (DAT-AR for PCA, [25]). The Abstract Reasoning (AR) scale of the DAT used in this experiment is included in the DAT-5 Spanish adaptation by the publisher TEA [26]. This test is usually used as a non-verbal measure of reasoning ability and involves the capacity to think logically and to perceive relationships in abstract figure patterns. It is considered as a marker of fluid intelligence [27], the component of intelligence most related to general intelligence or g factor [28]. The advantage of this test is that it is quite fast to implement: it consists of 40 multiple-choice items and has a 20-minute time limit. Subjects were informed that they would receive 0.25€ for each correct answer.

Most research investigating personality and giving in the context of the DG has focused on the common Big Five paradigm [12, 29–32, among others]. We elicit personality traits using the NEO-FFI-3 questionnaire proposed by McCrae & Costa [33] to control for the personality traits analyzed in the Big Five Model, i.e., neuroticism, extraversion, openness to experience, agreeableness, and conscientiousness [34]. The NEO-FFI-3 is a condensed version of other scales used to measure the Big Five Model, consisting of 60 items, 12 items for each personality trait.

Based on previous findings that emotions can influence dictator giving [35, 36], we also collected self-reported information on the main reason for the dictator's decision, offering two options: rationality or emotion. Subjects who selected the latter were asked to choose which emotion they felt the most as a basis of their decision, among the following: empathy, happiness, compassion, excitement, guilt, greed, fear, regret and "other". In our analysis, we aggregate emotions into "positive" (empathy, happiness, compassion, and excitement), "negative" (guilt, greed, fear, and regret), and "other". The list of emotions we used is the one proposed by Levine et al. [37]. Our measure does not rely on sophisticated technology, but it is simple and efficient in obtaining the self-reported motivation of the decision-maker. A possible limitation of this measure is that we cannot be sure whether the emotions subjects stated to have had were causing the decision or whether their reporting ex-post was affected by the decision made.

**Table 1. Descriptive statistics and definition of the main variables used in the analysis.**

| Variable | Definition | Mean (Standard Deviation) |
|---|---|---|
| Share sent (dependent variable) | Share of 10 euros sent by the dictator to the recipient, ranging from 0 to 1. | 0.31 (0.22) |
| Female | = 1 if female, = 0 otherwise | 0.58 |
| Age | Age. Ranging from 18 to 71. | 23.6 (0.21) |
| Reasoning Ability | Number of correct answers in the DAT-AR test. Ranges from 4 to 40. | 23.8 (0.21) |
| Neuroticism | Neuroticism scaled score on NEO-FFI, ranging from 18 to 77. | 47.1 (0.31) |
| Extraversion | Extraversion scaled score on NEO-FFI, ranging from 24 to 82. | 57.2 (0.29) |
| Openness | Openness scaled score on NEO-FFI, ranging from 26 to 79. | 53.9 (0.27) |
| Agreeableness | Agreeableness scaled score on NEO-FFI, ranging from 11 to 79. | 52.5 (0.30) |
| Conscientiousness | Conscientiousness scaled score on NEO-FFI, ranging from 15 to 71. | 50.5 (0.31) |
| Experience in experiments | Measured by means of a five-point Likert scale (0 = inexperienced, 1 = one to three sessions, 2 = four to 6 sessions, 3 = seven to nine sessions, 4 = more than nine sessions). Ranges from 0 to 4. | 1.19 (0.03) |
| Response time | Number of seconds between the appearance of the screen with the question on how much to send and the confirmation of the choice by the subject, ranging from 2 to 136. | 15.7 (0.33) |
| Rationality | = 1 if self-reported main motive for the decision was "rationality", = 0 otherwise | 0.79 |
| Positive emotion | = 1 if self-reported main motive for the decision was "Compassion", "Empathy", "Happiness" or "Excitement"; = 0 otherwise | 0.13 |
| Negative emotion | = 1 if self-reported main motive for the decision was "Guilt", "Regret", "Fear" or "Greed"; = 0 otherwise | 0.07 |
| "Other" emotion | = 1 of self-reported main motive for the decision was "Emotion—other", = 0 otherwise | 0.01 |

Total Number of Observations = 1,161.

The meta-analysis in [38] reveals that older adults show more altruistic behavior than younger adults. Moreover, having participated in laboratory experiments in the past may also influence dictator giving, as reported in [39]. These considerations led us to control for age and experience in lab experiments in our study.

Last, Cappelen et al. [40] find that there is a negative link between fair behavior and response time, suggesting that fairness is intuitive. Moreover, Rand et al. [41] observe increased DG giving in women when response time is exogenously reduced. We explore this possible link between response time and giving in our analysis.

## Results and discussion

### Descriptive analysis

Table 1 presents the descriptive statistics and definitions of the main variables used in the analysis. In line with previous studies (see [10] or [3]), dictators shared on average almost one third of their endowment. The average age was about 24 years old, and 58% of participants were female. The average number of correct answers in the DAT-AR test are similar to the ones calculated for the Spanish population [26]. The raw NEO-FFI test scores that we obtain are also in line with the means reported by McCrae and Costa [33] for university students.

Table 2 displays the mean and median shares sent in the Dictator Game by males and females, and Fig 1 shows the distribution of these shares by gender. Our main finding is that

**Table 2. Shares sent by dictators.**

| | Number of observations | Mean | Standard Deviation | Median | Mode | Percentage non-zero amounts sent |
|---|---|---|---|---|---|---|
| **Female** | 675 | 0.35 | 0.21 | 0.4 | 0.5 | 91.9% |
| **Male** | 486 | 0.25 | 0.23 | 0.23 | 0 | 74.5% |

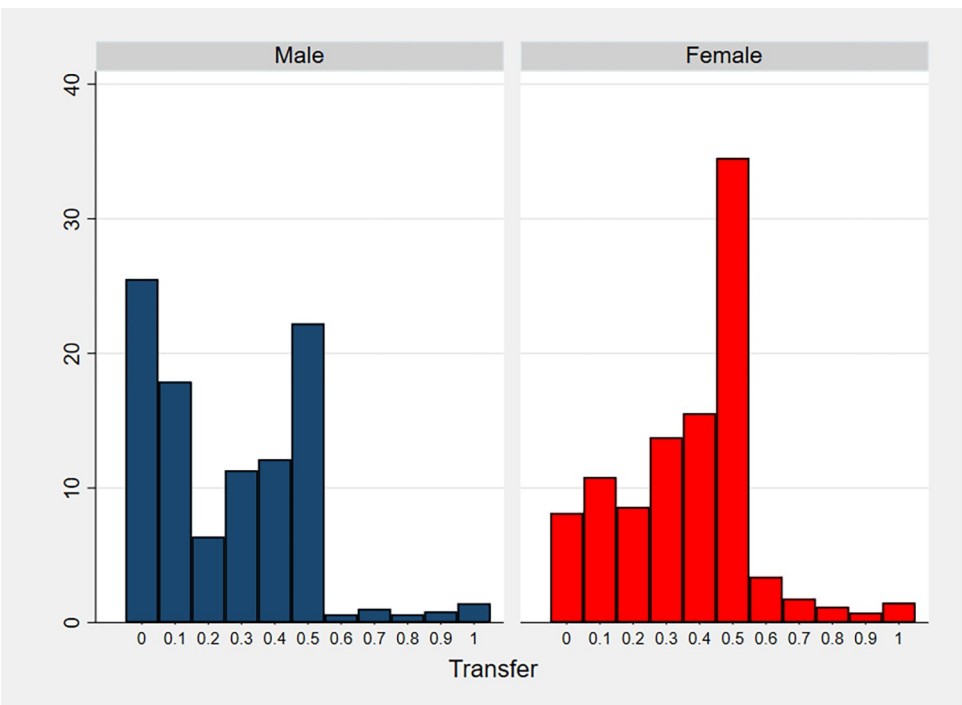

**Fig 1. Share of the endowment sent by dictators, distribution by gender.**

women transfer on average a higher fraction of their endowment (35%) than men (25%), and this difference is statistically significant. Using a one tailed t-test to confirm our pre-registered hypothesis, we obtain a t-statistic of –8.039, p-value < 0.001. We obtain the same result using a non-parametric Mann-Whitney test: z-statistic = -8.190, with p-value < 0.001.

Moreover, about 92% of women sent non-zero amounts, compared to 75% of men, and median shares were 40% for females, 23% for males. The most frequently transferred amount was 0 for men, while women's decisions were more concentrated around the "equalitarian" one-half share of the endowment, as in [1].

Fig 2 presents the distribution of the number of observed correct answers to the 40 multiple-choice items in the DAT-AR test, by gender. The mean and median number of right answers for females were both 22, with a standard deviation of 6.8. In contrast, males answered correctly on average to 25.7 items (standard deviation of 7.0, median of 26). Thus, reasoning ability scores were higher for men than women in our sample (Mann-Whitney test z-statistic = 7.693, p-value< 0.001). The correlation between the female and cognitive ability variables is significantly negative but small (Spearman correlation coefficient of -0.23, p-value < 0.001).

In Fig 3, we present a comparison of players' personality traits by gender as resulting from the NEO-FFI test scaled scores.

Women are found to be significantly more neurotic, open to experience, agreeable and conscientious than men (Mann-Whitney test z-statistics -2.915, -4.152, -3.034 and –4.890; p-values are 0.004, less than 0.001, 0.002, and less than 0.001 respectively). Neuroticism and agreeableness had already been found to be significantly greater in women in McCrae and Costa [33].

Slightly above three quarters of the dictators (391 of 486 males and 525 of 675 females) reported that their giving decision was based on rationality rather than emotion, as can be seen in the pie chart of Fig 4. For that figure, we grouped self-reported emotions into three categories: "positive" (empathy, happiness, compassion, and excitement), "negative" (guilt, greed,

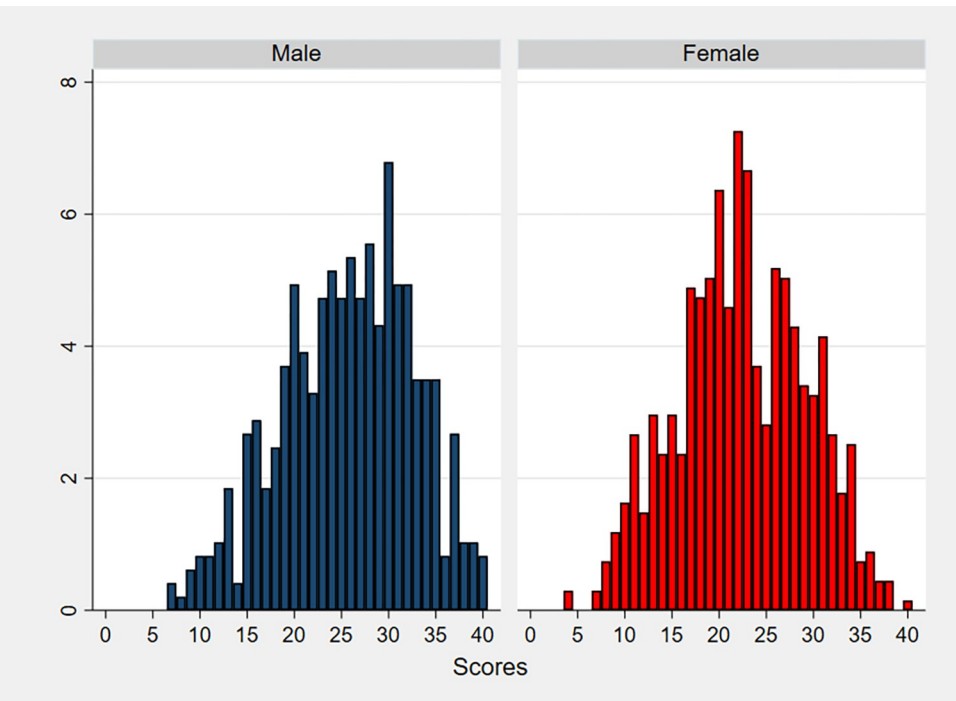

**Fig 2. Percentage of males and females per score in the DAT-AR test.**

fear, and regret) and "other" emotions. We observe that, of the quarter of subjects who stated that their choice was driven by some emotion, most people reported positive rather than negative emotions. Of those choosing a positive emotion, most indicated that they were driven by empathy (54% of males and 65% of females). On the other hand, greed was the most frequent negative emotion (87% of males and 70% of females).

"Positive emotion" dictators transferred higher amounts to recipients than "rationality"-driven individuals did, while "negative emotion" subjects gave lower amounts (Mann-Whitney tests show that these differences are significant: z-statistics of –4.520 and 5.993; Bonferroni-corrected p-values of less than 0.001 in both cases). The same result holds when the tests are run for females (the corresponding z-statistics are –3.224 and 5.292; p-values are 0.004 and < 0.001) and males separately (z-statistics –2.740 and 3.084; p-values of 0.018 and 0.006).

## Regression analysis

To finetune our descriptive and inferential results, we study now the effect on dictator giving of both gender and the other variables described in Table 1 by estimating the following OLS regression:

$$s_i = \beta_0 + \beta_1 Female_i + \beta_2 X_i + \beta_3 RT_i + \beta_4 M_i + \varepsilon_i$$

where $s_i$ indicates the share of the endowment transferred by dictator $i$, $Female_i$ is a dummy taking the value of 1 if the subject is a female, $X_i$ is the vector of individual characteristics, $RT_i$ is the response time measured in seconds, $M_i$ is a vector of dummies indicating the self-reported main motive for the decision, and $\varepsilon_i$ is the error term corresponding to individual $i$. Results are reported in Table 3. In the first column, we present the regression with the pre-registered variables (gender, reasoning ability, personality and emotions), while in the second column we also include age, experience in experiments and response time, which we routinely

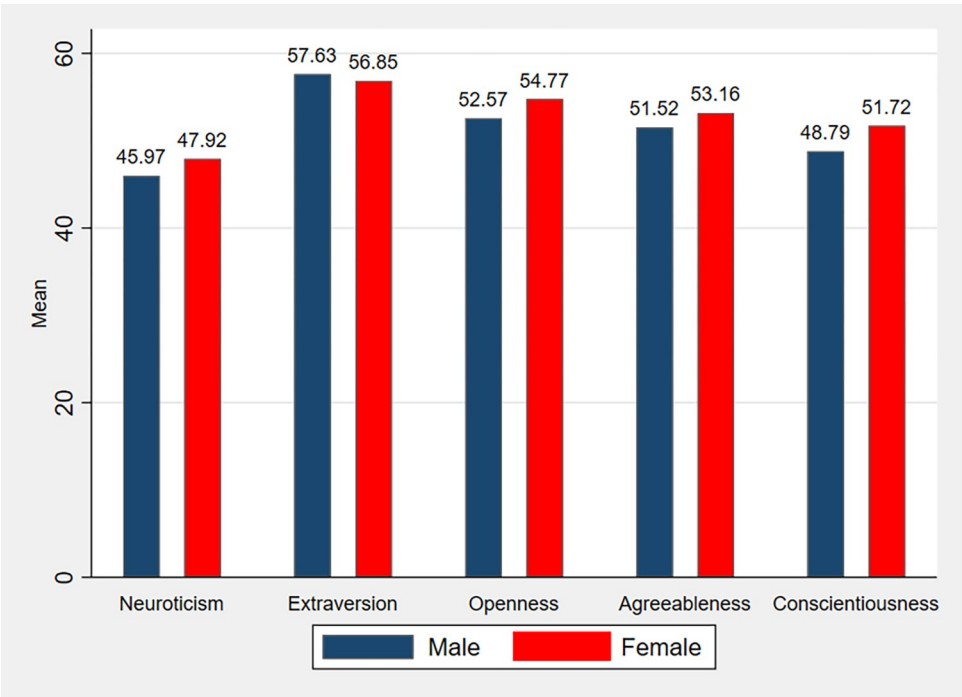

**Fig 3. Scaled scores in the NEO-Five factor inventory test, by gender.**

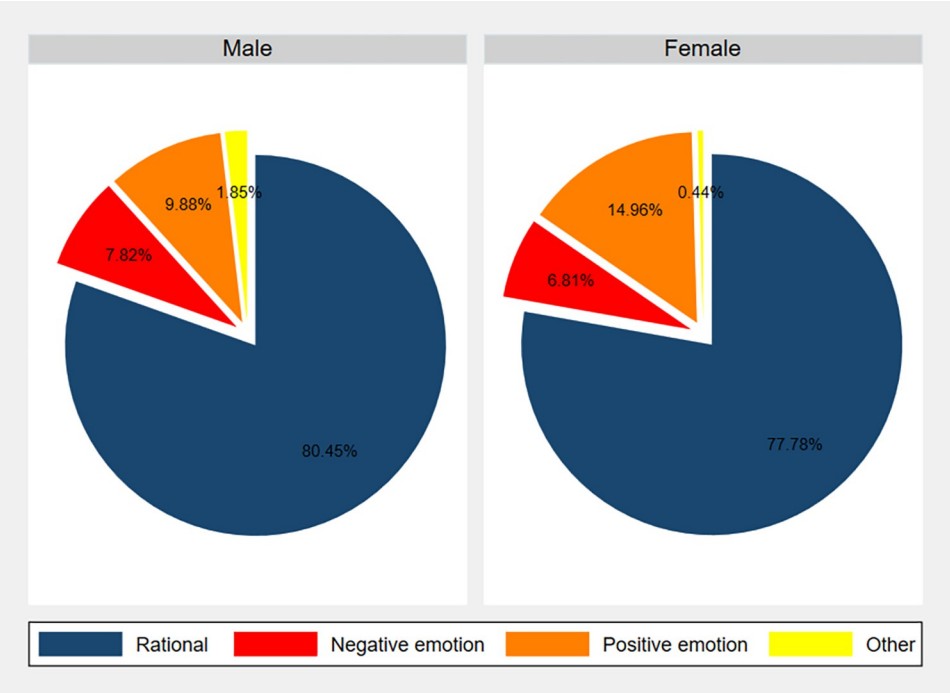

**Fig 4. Percentage of males and females reporting rationality vs. positive, negative, or other emotions as a reason for their decision in the DG.**

**Table 3. OLS regressions of the share transferred by dictators on gender, reasoning ability, personality traits, emotions, and other characteristics.**

| Dependent variable: Share sent | All Confirmatory (N = 1161) | All Exploratory (N = 1161) | Female (N = 675) | Male (N = 486) | Difference |
|---|---|---|---|---|---|
| | (1) | (2) | (3) | (4) | (5) = (3)—(4) |
| Constant | 0.1044 | 0.0572 | 0.2630** | -0.1530 | 0.4160* |
| | (0.0789) | (0.0819) | (0.0981) | (0.1523) | (0.1810) |
| Female | 0.0728*** | 0.0749*** | - | - | - |
| | (0.0136) | (0.0137) | | | |
| Reasoning Ability | -0.004*** | -0.0030** | -0.0032** | -0.0027 | -0.0005 |
| | (0.0009) | (0.0009) | (0.0011) | (0.0015) | (0.0019) |
| Neuroticism | 0.0010 | 0.0011 | 0.0010 | 0.0012 | -0.0002 |
| | (0.0007) | (0.0007) | (0.0008) | (0.0012) | (0.0014) |
| Extraversion | -0.0007 | -0.0004 | -0.0006 | -0.0000 | -0.0006 |
| | (0.0007) | (0.0007) | (0.0008) | (0.0013) | (0.0015) |
| Openness | 0.0021** | 0.0018** | 0.0015 | 0.0023* | -0.008 |
| | (0.0007) | (0.0007) | (0.0008) | (0.0012) | (0.0014) |
| Agreeableness | 0.0028*** | 0.0028*** | 0.0017* | 0.0045*** | -0.0028* |
| | (0.0006) | (0.0006) | (0.0007) | (0.0010) | (0.0013) |
| Conscientiousness | -0.0001 | -0.0005 | -0.0010 | 0.0005 | -0.0015 |
| | (0.0007) | (0.0006) | (0.0008) | (0.0011) | (0.0013) |
| Positive emotion | 0.0916*** | 0.0908*** | 0.0841*** | 0.1121** | -0.0280 |
| | (0.0191) | (0.0190) | (0.0216) | (0.0384) | (0.0439) |
| Negative emotion | -0.1461*** | -0.1296*** | -0.1405*** | -0.1161*** | -0.0244 |
| | (0.0161) | (0.0168) | (0.0229) | (0.0253) | (0.0341) |
| Other emotion | 0.0320 | 0.0407 | -0.0089 | 0.0730 | -0.0819 |
| | (0.0677) | (0.0708) | (0.1802) | (0.0766) | (0.1960) |
| Age | - | 0.0028** | 0.0024 | 0.0034* | -0.0010 |
| | | (0.0010) | (0.0015) | (0.0013) | (0.0020) |
| Experience in experiments | - | -0.0301*** | -0.0233** | -0.0365*** | 0.0133 |
| | | (0.0054) | (0.0072) | (0.0082) | (0.0109) |
| Response time | - | -0.0004 | -0.0007 | 0.0002 | -0.0009 |
| | | (0.0007) | (0.0008) | (0.0010) | (0.0013) |
| R-squared | 0.149 | 0.174 | 0.113 | 0.164 | - |

Robust standard errors of estimated coefficients in parentheses. Significance levels are marked with

*** for p-values < 0.001

** for p-values < 0.01, and

* for p-values < 0.05.

collect in our lab, but did not explicitly mention in the pre-registration. The results remain qualitatively unchanged. In columns 3 and 4, we present the coefficients estimated in separate regressions for females and males respectively, and their difference in column 5. The coefficients in this last column have been estimated by interacting each independent variable with the female dummy to test for any significant moderating effects.

Confirming our hypothesis and the inferential analysis reported above, we obtain that female dictators share on average 7.5 percentage points more than men, being this difference highly statistically significant. After controlling for the effect of idiosyncratic variables on female and male decisions separately, women transfer on average 41.6% more than men. This difference that we find between male and female giving is a bit higher than those reported in the meta-analyses by Engel [10], Bilén et al. [3] or Doñate-Buendía et al. [15]. This could be due to cultural differences (Doñate-Buendía et al. [15] find that gender differences in DG vary across locations), or maybe women show a more empathic response towards their recipients in a dual-role version of the DG.

Apart from gender, we find that reasoning ability, two personality traits, emotions, age, and experience in experiments are also significant predictors of dictator's giving. A negative correlation between reasoning ability and giving behavior in the DG has previously been obtained under different cognitive tests: the Wonderlic [12], the Raven test [42], the Cognitive Reflection Test [43], the Mac Game [44]. In line with these results, we also find a negative correlation between cognitive ability and DG giving, even if this effect is quite small and significant only for women.

In our case, the only personality traits that seem to play a role in dictator giving are openness and agreeableness. Specifically, male dictators that are more "open to experience" transfer more to their recipients, while "agreeableness" increases generosity for both males and females, but to a different extent. That is, the relationship between this personality trait and dictator giving is moderated by gender: agreeableness increases giving significantly more for males than for females.

A positive effect of openness on DG giving had already been found in [12], while several studies find that agreeableness is associated with greater DG giving ([30, 31], among others; see also the review in [32]).

Moreover, we find that, independently of their gender, subjects who report a positive emotion as their main motivation for sharing transfer a higher amount to recipients than those who declare that their decision was based on rationality. On the other hand, negative emotions negatively affect dictators' sharing. This effect of emotions is quite big and significant. Our findings are consistent with those in Fiala and Noussair [35] that subjects in a more positive overall emotional state tend to donate more. In particular, empathy, the most frequently reported positive emotion in our experiment, has been found to be a strong predictor of sharing in the DG game [36].

Supporting results obtained in Sparrow et al. [38] meta-analysis, we find that altruism increases with age. However, this age effect is significant only for male dictators. This might be because most of our subjects are in their twenties and very few are over forty years old.

Previous experience in experiments has a strong negative effect on dictators' transfers. This is in line with Tjøtta [39]'s result that participants' experience in experiments is negatively correlated with splitting fairly in the dictator game.

Last, in our case, the time employed to implement the dictator's decision does not affect altruism. This is in contrast with Cappelen et al. [40], who find that the response time of the dictators that give half of the endowment is significantly lower than the response time of the dictators who give nothing. Even focusing on this special case, we do not replicate their results (Mann-Witney test z-statistics –0.083, -0.077 and –0.735; p-values are 0.9342, 0.9387, and 0.4626, for all data, males, and females, respectively).

## Robustness analysis: Structural equations modeling

As a robustness analysis and to test whether the gender effect on dictator giving is moderated by different individual characteristics and emotions between men and women, we follow a Structural Equation Modeling (SEM) approach. We used the command "sem" in the STATA 18 software. Fig 5 shows the path diagram of the structural model and fit statistics, together with the estimated parameters and corresponding standard errors and p-values (significant estimates are marked in bold). The figure shows the direct and indirect effects of gender on the share sent in the Dictator Game, controlling for age and previous experience in experiments.

Table 4 presents the detailed regression results, together with direct and indirect effects. As for the former, our results are very robust compared to the ones obtained in the regression analysis presented in the second column of Table 3. Additionally, we find that the effect of

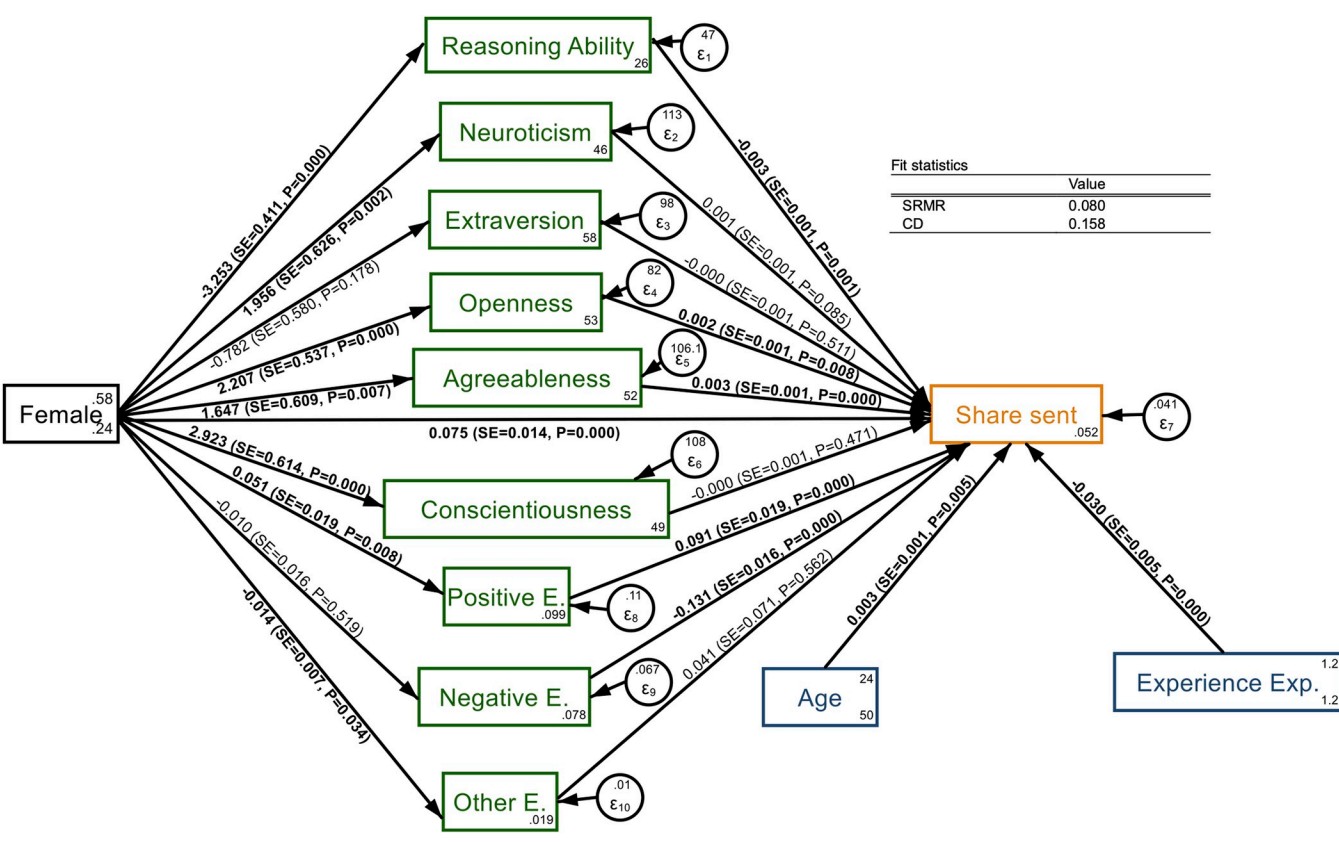

**Fig 5. Path diagram of the SEM for the share transferred by dictators, with estimated coefficients and fit statistics.**

gender is mediated by reasoning ability, openness, agreeableness, and self-reported positive emotions. The net indirect effect of gender mediated by these variables is estimated to be around 2.5 percentage points, positive and significant (p-value = 0.0006). However, the gender effect is not fully mediated by these factors, as the coefficient on Female remains significant: the estimated direct effect is that female subjects share 7.5 percentage points more than men (p-value < 0.0001). Thus, the total effect resulting from the SEM estimation is that women give on average 10 percentage points more than men, in line with the descriptive and inferential analysis.

The SEM model has an acceptable fit as the standardized root mean squared residual is 0.08. The error variance of our dependent variable is around 0.04, with a standard error of 0.002. The fraction of variance explained by our model (coefficient of determination) is about 16%. This relatively low value could be due to omitted explanatory variables like important mediators of gender differences in other contexts, such as Social Value Orientation [45], or other factors that might affect giving, as inequality aversion [46, 47] or warm glow [48].

## Conclusions

We collected a large dataset on transfers in a dictator game together with other individual characteristics, which allowed us to study gender differences in DG giving controlling for a range of factors potentially affecting this decision. We contribute to solving the problem of lack of sufficient power in previous laboratory tests of gender differences in DG giving [3], generating robust evidence to elucidate previous inconclusive results [2]. Our t-test has an ex-

**Table 4. Mediation analysis with structural equation modeling.**

| | Reasoning Ability | Neuroticism | Extraversion | Openness | Agreeableness | Conscientiousness | Positive emotion | Negative emotion | Other emotion | Share Sent Direct effects | Indirect effects |
|---|---|---|---|---|---|---|---|---|---|---|---|
| Constant | 25.6605*** | 45.9671*** | 57.6296*** | 52.5669*** | 51.5160*** | 48.7922*** | 0.0988** | 0.0782*** | 0.0185*** | 0.0519 | |
| | (0.317) | (0.466) | (0.422) | (0.402) | (0.455) | (0.459) | (0.014) | (0.012) | (0.006) | (0.081) | |
| Female | -3.2531*** | 1.9559** | -0.7820 | 2.2069*** | 1.6467** | 2.9232*** | 0.0509*** | -0.0100 | -0.0141* | 0.0755*** | 0.0248*** |
| | (0.411) | (0.626) | (0.580) | (0.537) | (0.609) | (0.614) | (0.019) | (0.016) | (0.007) | (0.014) | (0.006) |
| Reasoning Ability | | | | | | | | | | -0.0030*** | |
| | | | | | | | | | | (0.001) | |
| Neuroticism | | | | | | | | | | 0.0011 | |
| | | | | | | | | | | (0.001) | |
| Extraversion | | | | | | | | | | -0.0004 | |
| | | | | | | | | | | (0.001) | |
| Openness | | | | | | | | | | 0.0018** | |
| | | | | | | | | | | (0.001) | |
| Agreeableness | | | | | | | | | | 0.0028*** | |
| | | | | | | | | | | (0.001) | |
| Conscientiousness | | | | | | | | | | -0.0005 | |
| | | | | | | | | | | (0.001) | |
| Positive emotion | | | | | | | | | | 0.0909*** | |
| | | | | | | | | | | (0.019) | |
| Negative emotion | | | | | | | | | | -0.1313*** | |
| | | | | | | | | | | (0.016) | |
| Other emotion | | | | | | | | | | 0.0410 | |
| | | | | | | | | | | (0.071) | |
| Age | | | | | | | | | | 0.0028** | |
| | | | | | | | | | | (0.001) | |
| Experience in experiments | | | | | | | | | | -0.0296*** | |
| | | | | | | | | | | (0.005) | |
| var(e.share_sent)| | | | | | | | | | | 0.0412 | |
| | | | | | | | | | | (0.002) | |

N = 1161. Robust standard errors of estimated coefficients in parentheses.

***p<0.001

**p<0.01

*p<0.05. var(e.share_sent) is the variance of the error in Share Sent.

ante power of 0.80 for a significance level of 0.05, obtaining that women give on average 10 percentage points (1 euro out of 10) more than men. When controlling for the set of potentially mediating variables, this difference is still 7.5 percentage points (about 40% higher average giving for females), and highly significant. This result is robust to using an Ordinary Least Squares Regression and a Structural Equation Modelling approach. Additionally, we find that gender differences in giving are mediated by reasoning ability, openness, agreeableness, and self-reported positive emotions.

The consequences of our findings are manifold. Economists' models, experiments and empirical analyses should consider these systematic gender differences in behavior whenever altruism is a relevant factor affecting decisions. This could be the case in charitable giving, volunteering, blood and organ donation, caregiving, and other intra-household decisions, but also in broader social contexts, such as politics (e.g., supporting redistributive policies), the environment (mobilizing collective action), or the workplace (collaborating with colleagues, etc.).

Our main limitation is that we have studied altruism in the economic domain, as willingness to give money. Moreover, our findings stem from a standard dual-role Dictator Game with a price of giving equal to one. Field experiments with large samples would be particularly useful to extrapolate these findings to other domains, given that differences in altruism are found to be context-specific [1, 49, 50]. Also, representative samples would increase the external validity of our results. Last, further avenues of research could explore the impact of gender differences in altruism on pro-social behavior (as cooperation, contributions to public goods, or trust), considering the need for large samples and control of idiosyncratic factors.

## Supporting information

**S1 File. Experimental instructions.**
(DOCX)

## Acknowledgments

The authors wish to acknowledge two anonymous reviewers and Holger Rau, the editor of this article, for their valuable comments.

## Author Contributions

**Conceptualization:** Iván Barreda-Tarrazona, Ainhoa Jaramillo-Gutiérrez, Marina Pavan, Gerardo Sabater-Grande.

**Data curation:** Ainhoa Jaramillo-Gutiérrez.

**Formal analysis:** Ainhoa Jaramillo-Gutiérrez.

**Funding acquisition:** Iván Barreda-Tarrazona, Gerardo Sabater-Grande.

**Investigation:** Iván Barreda-Tarrazona, Ainhoa Jaramillo-Gutiérrez, Marina Pavan, Gerardo Sabater-Grande.

**Project administration:** Iván Barreda-Tarrazona, Gerardo Sabater-Grande.

**Software:** Iván Barreda-Tarrazona.

**Supervision:** Gerardo Sabater-Grande.

**Visualization:** Ainhoa Jaramillo-Gutiérrez.

**Writing – original draft:** Iván Barreda-Tarrazona, Marina Pavan, Gerardo Sabater-Grande.

**Writing – review & editing:** Marina Pavan.

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
