## [Decision Letter · Decision Letter 0]

2 Jun 2024

PONE-D-24-05752Gender differences in dictator giving: a high-power laboratory testPLOS ONE

Dear Dr. Pavan,

Thank you for submitting your manuscript to PLOS ONE. After careful consideration, we feel that it has merit but does not fully meet PLOS ONE’s publication criteria as it currently stands. Therefore, we invite you to submit a revised version of the manuscript that addresses the points raised during the review process.

Dear Marina,

thank you for submitting your work to PLOS ONE. I now received feedback from two reviewers who are experts in their fields. As you can see from below, both reviewers like your work. While reviewer 2 thinks more positively about the paper, reviewer 1 criticizes the work more, but at the same time makes a lot of very useful comments that will improve the manuscript. From my point of view, the most important aspects are that you better have to motivate your contributions in connection with the already large literature on gender differences in altruism (e.g., Eckel & Grossman, 1996; Falk et al. 2018). With respect to Falk et al. (2018) who have a very large data set, it will be important to better motivate the contributions of your current work. This point is raised by reviewer #1, but also reviewer #2 requests a better classification of the study in the gender literature and related discussion. One way to follow will be a better motivation of other features of the paper, such as the presentation of potential mediating and moderating factors of giving, which is clearly interesting. This brings me to the next important point. I agree with reviewer #1 that you have to conduct a suitable mediation analysis, i.e., a Structural Equation Modeling. Moreover, I agree that at this point, you will have to provide a clear reasoning for how control variables were chosen, and you should discuss the different types of control variables. Moreover, for the revision, I expect you to reply to all comments of the reviewers, and if not possible to fix, you should clearly argue why this is not possible.

I am looking forward to the revised version of your manuscript.

Best

We look forward to receiving your revised manuscript.

Kind regards,

Holger A. Rau

Academic Editor

PLOS ONE

Journal Requirements:

2. Thank you for stating the following financial disclosure: "This work was supported by the Spanish Ministry of Science and Innovation (PID2021-123053OB-I00) and Jaume I University (UJI-B2021-23)."

3. Thank you for stating the following in the Acknowledgments Section of your manuscript: "This work was supported by the Spanish Ministry of Science and Innovation (PID2021-123053OB-I00) and Jaume I University (UJI-B2021-23)."

Please remove any funding-related text from the manuscript and let us know how you would like to update your Funding Statement. Currently, your Funding Statement reads as follows: "This work was supported by the Spanish Ministry of Science and Innovation (PID2021-123053OB-I00) and Jaume I University (UJI-B2021-23)."

Additional Editor Comments:

Dear Marina,

thank you for submitting your work to PLOS ONE. I now received feedback from two reviewers who are experts in their fields. As you can see from below, both reviewers like your work. While reviewer 2 thinks more positively about the paper, reviewer 1 criticizes the work more, but at the same time makes a lot of very useful comments that will improve the manuscript. From my point of view, the most important aspects are that you better have to motivate your contributions in connection with the already large literature on gender differences in altruism (e.g., Eckel & Grossman, 1996; Falk et al. 2018). With respect to Falk et al. (2018) who have a very large data set, it will be important to better motivate the contributions of your current work. This point is raised by reviewer #1, but also reviewer #2 requests a better classification of the study in the gender literature and related discussion. One way to follow will be a better motivation of other features of the paper, such as the presentation of potential mediating and moderating factors of giving, which is clearly interesting. This brings me to the next important point. I agree with reviewer #1 that you have to conduct a suitable mediation analysis, i.e., a Structural Equation Modeling. Moreover, I agree that at this point, you will have to provide a clear reasoning for how control variables were chosen, and you should discuss the different types of control variables. Moreover, for the revision, I expect you to reply to all comments of the reviewers, and if not possible to fix, you should clearly argue why this is not possible.

I am looking forward to the revised version of your manuscript.

Best

Holger

Reviewers' comments:

Reviewer's Responses to Questions

**Comments to the Author**

1. Is the manuscript technically sound, and do the data support the conclusions?

Reviewer #1: Partly

Reviewer #2: Yes

2. Has the statistical analysis been performed appropriately and rigorously? 

Reviewer #1: No

Reviewer #2: Yes

3. Have the authors made all data underlying the findings in their manuscript fully available?

Reviewer #1: Yes

Reviewer #2: Yes

4. Is the manuscript presented in an intelligible fashion and written in standard English?

Reviewer #1: Yes

Reviewer #2: Yes

5. Review Comments to the Author

Reviewer #1: Summary: The paper investigates gender differences in dictator giving based on the decisions of 1161 participants (students). Women give a significantly higher fraction of their endowment (35%) to an unknown co-participant than men (25%). The size of this gender difference is comparable to recent meta-studies. In addition, the paper considers potential mediating and moderating factors of these gender differences. Precisely, they control for age, cognitive ability, personality traits, response time, and self-reported emotions motivating the decision. They find that age, cognitive ability, experience in experiments, openness and agreeableness as well as emotions are positively correlated with giving. In addition, agreeableness has been identified as a moderating factor.

General assessment: The paper contributes to previous literature by investigating gender differences in dictator giving using a large sample. It is also transparent in how the data was collected and based on a pre-registered analysis plan. While I appreciate these features, the paper comes short in explaining why this exercise is important besides referring to the Meta-Analysis by Bilén et al. (2021), showing that other studies are underpowered. The authors claim that they add to the question of gender differences in altruism but fail to explain how they add compared to important studies like Falk et al. (2018), who have a substantially larger sample (80 000 people) and find that women are more altruistic than men using a survey measure that has been validated using the dictator game. The difference to this paper is that the current paper uses dictator giving as a proxy for altruism and collects data in the lab. However, why is this an advantage? Especially given that previous literature shows that gender differences in giving in the dictator game are highly dependent on the exact design (e.g., is the gender of the recipient known or is it a double-blind design, are there efficiency concerns, is the money given to a charity…), it is not clear what we can learn from observing a significant gender difference in this specific example. Another potential contribution of the paper is the investigation of other factors like personality traits, etc., that affect giving and could potentially mediate and moderate gender differences. While I find this aspect very interesting, the paper does not provide a proper mediation analysis to help us understand which factors mediate the gender difference. Also, the variables added seem to be a mixture of mediator variables (personality traits), control variables (age, cognitive ability), and variables that might be endogenous to the decision of how much to give rather than affecting the decision (emotions, response time). There is no consistent theoretical framework motivating the choice of variables that are included, nor is there a distinction between the different types of variables. Below, I provide some comments that hopefully help improve the paper further.

Major comments:

Contribution: While I agree that the paper adds to the literature on dictator giving, it is less clear how it adds to the question of gender differences in altruism, given that Falk et al. (2018) find gender differences in altruism in a much larger sample. I think the paper would benefit from a clear discussion of this paper and why having a lab study is important. In addition, there should be a discussion of the limitations that come with only looking at one specific version of the Dictator Game. See also Exley et al. (2023) for a recent discussion of the importance of context for the emergence of gender differences in social preferences.

"Control Variables": I think including additional variables is an interesting extension to the previous literature. Unfortunately, the paper does not provide a proper mediation analysis (SEM) to help us understand which factors mediate the gender difference. It could be interesting to present the share of the gender difference they are mediating. Also, it would be helpful to provide a clear reasoning for how control variables were chosen. Right now, it seems a bit random and lacks a theoretical foundation. In addition, the variables added seem to be a mixture of mediator variables (personality traits), control variables (e.g., age), and variables that might be endogenous to the decision of how much to give rather than affecting the decision (e.g., emotions, response time). I think it would increase the paper's clarity to clearly distinguish between the different types of "control" variables added and discuss them in a more structured way.

Clarity concerning tests: Throughout the paper, it is not stated clearly whether one- or two-sided tests are reported, whether there is a correction for multiple hypotheses testing, and if yes, which hypotheses were grouped together.

Ex-post power calculation: There is no benefit in conducting the ex-post power calculation. In fact, using the estimated treatment effect for ex-post power calculations is misleading. See, for example, the discussion at https://blogs.worldbank.org/impactevaluations/why-ex-post-power-using-estimated-effect-sizes-bad-ex-post-mde-not

Exploratory analysis: In the pre-registration, the authors state that they "will also run an OLS regression of the dictator giving variable on gender, controlling for cognitive ability, personality, and emotions." However, in the paper, they also include age and response time. The exploratory part of the analysis should be marked clearly.

Minor Comments:

Tables: It would be helpful to include the number of observations in the Tables.

Instructions: I could not find the full instructions for the experiment. How was the rationality/emotion question asked?

References not included in the paper:

Exley et al. (2022) Harvard Business School Working Paper 22-079 https://www.hbs.edu/ris/Publication%20Files/22-079_8fd9dfda-2985-4192-84f8-e12748063b3b.pdf

Reviewer #2: Referee report on Gender differences in dictator giving: a high-power laboratory test (Manuscript Number: PONE-D-24-05752)

This study investigates the presence of gender differences through lab experiments in the canonical game of altruism, the dictator game. As the authors note, the high number of observations allows for drawing firm conclusions in a topic on which there is not yet consensus. The authors document a significant gender difference, finding that females are more generous than males, even after accounting for a wide range of relevant control variables such as age, cognitive ability, emotions and personality traits.

This study is of high quality and makes a clear contribution to the literature, making it deserving of publication. However, there are some minor issues that should be addressed beforehand.

Comments

I did not understand how playing the dictator game in dual roles affects the results. The authors emphasize the relevance of dual roles, but based on the description of the experiment, I did not see why this is important. Perhaps this should be made clearer.

I appreciate that this paper contributes to the debate on whether women are more generous than men. Although many studies, and perhaps even most, document a significant difference, the question remains unsettled due to the low number of observations and some influential papers (e.g., Niederle, reference 2 in the study). The authors make a strong argument for the presence of such a gender difference. Additionally, there are some recent studies (e.g., Brañas-Garza 2024, Horn et al. 2022) that use large samples to test gender differences in social preferences. This field is progressing rapidly, with studies using increasingly large sample sizes. Therefore, I suggest that the authors review the related literature from recent years to ensure that all relevant studies are included. It may also be interesting to see if these new studies with large sample sizes consistently point in the same direction.

In some cases, the authors use non-parametric tests, while in others, they use parametric ones. Using non-parametric tests consistently, which do not require a normality assumption, would make the paper more consistent and slightly more convincing.

I really liked the emotion questions. I have the impression that the authors should give them more emphasis, as they seem to be novel, easy to carry out, and potentially relevant for understanding social preferences.

References

Brañas-Garza, P. (2024). Young teens at play: Girls are egalitarian, boys are generous. Personality and Individual Differences, 226, 112703.

Horn, D., Kiss, H. J., & Lénárd, T. (2022). Gender differences in preferences of adolescents: evidence from a large-scale classroom experiment. Journal of Economic Behavior & Organization, 194, 478-522.

6. PLOS authors have the option to publish the peer review history of their article (what does this mean?). If published, this will include your full peer review and any attached files.

Reviewer #1: No

Reviewer #2: No

---

## [Author Response · Author response to Decision Letter 0]

4 Oct 2024

Response to the Editor and Reviewers PONE-D-24-05752

Gender differences in dictator giving: a high-power laboratory test

Dear Marina,

thank you for submitting your work to PLOS ONE. I now received feedback from two reviewers who are experts in their fields. As you can see from below, both reviewers like your work. While reviewer 2 thinks more positively about the paper, reviewer 1 criticizes the work more, but at the same time makes a lot of very useful comments that will improve the manuscript. From my point of view, the most important aspects are that you better have to motivate your contributions in connection with the already large literature on gender differences in altruism (e.g., Eckel & Grossman, 1996; Falk et al. 2018). With respect to Falk et al. (2018) who have a very large data set, it will be important to better motivate the contributions of your current work. This point is raised by reviewer #1, but also reviewer #2 requests a better classification of the study in the gender literature and related discussion. One way to follow will be a better motivation of other features of the paper, such as the presentation of potential mediating and moderating factors of giving, which is clearly interesting. This brings me to the next important point. I agree with reviewer #1 that you have to conduct a suitable mediation analysis, i.e., a Structural Equation Modeling. Moreover, I agree that at this point, you will have to provide a clear reasoning for how control variables were chosen, and you should discuss the different types of control variables. Moreover, for the revision, I expect you to reply to all comments of the reviewers, and if not possible to fix, you should clearly argue why this is not possible.

I am looking forward to the revised version of your manuscript.

Best

Authors: Dear Editor, many thanks for giving us the opportunity to revise our work in the light of the constructive comments of the reviewers. In the revised version that we submit, we have connected our laboratory study to the relevant papers on the topic that use a non-laboratory methodology (e.g., Falk et al., 2018). We have linked the choice of control variables to the previous findings in the literature which point to their potential relevance in explaining dictator giving. We have also included a robustness check of our regression analysis estimating a Structural Equations Model, which confirms the significant moderating effect of gender on the share sent by dictators.

We answer in detail to each reviewer’s comments below. We also have revised the manuscript in order to comply with the style requirements of Plos ONE.

Authors’ response to Reviewer 1’s comments:

Authors: We thank you for the constructive comments, which have helped us improve the paper. Here are our responses. Your original comments are included for convenience, and our reply is below.

5. Review Comments to the Author

Reviewer #1: Summary: The paper investigates gender differences in dictator giving based on the decisions of 1161 participants (students). Women give a significantly higher fraction of their endowment (35%) to an unknown co-participant than men (25%). The size of this gender difference is comparable to recent meta-studies. In addition, the paper considers potential mediating and moderating factors of these gender differences. Precisely, they control for age, cognitive ability, personality traits, response time, and self-reported emotions motivating the decision. They find that age, cognitive ability, experience in experiments, openness and agreeableness as well as emotions are positively correlated with giving. In addition, agreeableness has been identified as a moderating factor.

General assessment: The paper contributes to previous literature by investigating gender differences in dictator giving using a large sample. It is also transparent in how the data was collected and based on a pre-registered analysis plan. While I appreciate these features, the paper comes short in explaining why this exercise is important besides referring to the Meta-Analysis by Bilén et al. (2021), showing that other studies are underpowered. The authors claim that they add to the question of gender differences in altruism but fail to explain how they add compared to important studies like Falk et al. (2018), who have a substantially larger sample (80 000 people) and find that women are more altruistic than men using a survey measure that has been validated using the dictator game.

The difference to this paper is that the current paper uses dictator giving as a proxy for altruism and collects data in the lab. However, why is this an advantage? Especially given that previous literature shows that gender differences in giving in the dictator game are highly dependent on the exact design (e.g., is the gender of the recipient known or is it a double-blind design, are there efficiency concerns, is the money given to a charity…), it is not clear what we can learn from observing a significant gender difference in this specific example. Another potential contribution of the paper is the investigation of other factors like personality traits, etc., that affect giving and could potentially mediate and moderate gender differences. While I find this aspect very interesting, the paper does not provide a proper mediation analysis to help us understand which factors mediate the gender difference. Also, the variables added seem to be a mixture of mediator variables (personality traits), control variables (age, cognitive ability), and variables that might be endogenous to the decision of how much to give rather than affecting the decision (emotions, response time). There is no consistent theoretical framework motivating the choice of variables that are included, nor is there a distinction between the different types of variables. Below, I provide some comments that hopefully help improve the paper further.

Contribution: While I agree that the paper adds to the literature on dictator giving, it is less clear how it adds to the question of gender differences in altruism, given that Falk et al. (2018) find gender differences in altruism in a much larger sample. I think the paper would benefit from a clear discussion of this paper and why having a lab study is important. In addition, there should be a discussion of the limitations that come with only looking at one specific version of the Dictator Game. See also Exley et al. (2023) for a recent discussion of the importance of context for the emergence of gender differences in social preferences.

Authors: We have included in the introduction the following sentence on the relevance of conducting a laboratory experiment:

“Using the methodology of experimental economics to conduct well incentivized and properly controlled laboratory experiments greatly facilitates the identification of causal relationships in comparison to other approaches (as surveys, online or field experiments).”

Also, in the introduction, we now discuss a series of papers that study gender differences in Dictator Giving and that involve large sample sizes but are not laboratory experiments, including Falk et al. (2018). All these studies present lower control than lab experiments or reduced incentives, or both. 

Following your suggestion, we discuss Falk et al. (2018) as follows:

“The Global Preference Survey presented in Falk et al. (2018) and collecting data from 80,000 participants in 76 countries also contains two altruism related items: a hypothetical charity donation decision and a self-assessment of the willingness to give to good causes. They find that altruism is more pronounced among women. The survey questions were chosen based on their explanatory power of the decisions of 409 experimental subjects participating in three binary Charity Dictator Games, but the incentives at stake amounted to only 2.4 euros.”

We have added a paragraph in the Experimental Design and Methods section mentioning the two possible differential effects of using a dual role version of the dictator game with respect to the single role one, namely an increased empathy towards the counterpart and a decreased responsibility regarding the final payoffs. However, our goal in this study was not finding out which of these two likely effects prevailed in our sample, but just testing for gender differences in dictator giving keeping the game implementation invariant.

In the conclusions we acknowledge the limitation that the differences in Dictator Giving that we find could be context-specific, adding also the reference of Exley et al. (forthcoming) that you suggested. The paragraph now reads as follows:

“Our main limitation is that we have studied altruism in the economic domain, as willingness to give money. Moreover, our findings stem from a standard dual-role Dictator Game with a price of giving equal to one. Field experiments with large samples would be particularly useful to extrapolate these findings to other domains, given that differences in altruism are found to be context-specific (Andreoni and Vesterlund, 2001; Eckel and Grossman, 1996; Exley et al., forthcoming).”

"Control Variables": I think including additional variables is an interesting extension to the previous literature. Unfortunately, the paper does not provide a proper mediation analysis (SEM) to help us understand which factors mediate the gender difference. It could be interesting to present the share of the gender difference they are mediating. Also, it would be helpful to provide a clear reasoning for how control variables were chosen. Right now, it seems a bit random and lacks a theoretical foundation. In addition, the variables added seem to be a mixture of mediator variables (personality traits), control variables (e.g., age), and variables that might be endogenous to the decision of how much to give rather than affecting the decision (e.g., emotions, response time). I think it would increase the paper's clarity to clearly distinguish between the different types of "control" variables added and discuss them in a more structured way.

Authors: We present and justify the inclusion of each independent variable used in the analysis in the subsection “Assessing reasoning ability, personality traits and emotions” in the Experimental Design and Methods section. The choice of each of these variables is driven mainly by the findings of the extensive previous literature on dictator giving.

We also added the sub-section “Robustness Analysis: Structural Equation Modeling” in which we present the path diagram of a SEM with the estimated coefficients and fit statistics, indirect and total effects of the share transferred by dictators, and the test for gender invariance of the parameters.

In that model, the gender dummy is the moderator variable, while personality traits, emotions, and reasoning ability mediate the relationship between age and our dependent variable. On the other hand, reasoning ability mediates the effect of lab experience on dictator giving. To correct for potential endogeneity, we include in the model the covariance between response time and the share sent. However, we consider emotions as an exogenous mediating variable, as the question we asked was about the main reason for the subject’s decision, offering two options: rationality or emotion. If they answered the latter, they were asked to choose which emotion they felt the most as a basis for their decision. We did not ask how the subjects felt because of the decision (which would be certainly endogenous).

As you can see, the results from the SEM analysis are very robust compared to the ones obtained in the original regression analysis.

Clarity concerning tests: Throughout the paper, it is not stated clearly whether one- or two-sided tests are reported, whether there is a correction for multiple hypotheses testing, and if yes, which hypotheses were grouped together.

Authors: Thank you for pointing out this lack of precision. We now better specify whether a test is one-sided and whether we have Bonferroni-corrected for multiple comparisons.

Ex-post power calculation: There is no benefit in conducting the ex-post power calculation. In fact, using the estimated treatment effect for ex-post power calculations is misleading. See, for example, the discussion at https://blogs.worldbank.org/impactevaluations/why-ex-post-power-using-estimated-effect-sizes-bad-ex-post-mde-not

Authors: we followed your suggestions and deleted the ex-post power calculations in the manuscript.

Exploratory analysis: In the pre-registration, the authors state that they "will also run an OLS regression of the dictator giving variable on gender, controlling for cognitive ability, personality, and emotions." However, in the paper, they also include age and response time. The exploratory part of the analysis should be marked clearly.

Authors: It is true that in the pre-registration we do not include age, experience in experiments and response time as control variables. The first two are quite standard control variables that we routinely record in each experiment in our lab, while the third one is found to have a link with fair behavior in DG in the paper by Cappelen et al. (2016). In the presentation of our main results in Table 3, we now present a first column in which we do not include these three variables in the regression (so that this first column is totally consistent with our pre-registration statement). As you can see, the results are robust to the inclusion of the additional variables. 

Minor Comments:

Tables: It would be helpful to include the number of observations in the Tables.

Authors: We have made sure that the number of observations (indicated as N=#) is shown in every Table.

Instructions: I could not find the full instructions for the experiment. How was the rationality/emotion question asked?

Authors: Experimental Instructions can be found in the supporting information, submitted alongside the manuscript. The Rationality vs. Emotion as motive of the dictator giving decision questions were asked as described in the last paragraph of the Experimental Design and Methods Section: 

“We also collected self-reported information on the main reason for the dictator’s decision, offering two options: rationality or emotion. Subjects who selected the latter were asked to choose which emotion they felt the most when making their decision, among the following: empathy, happiness, compassion, excitement, guilt, greed, fear, regret and ‘other’.”

Authors’ response to Reviewer 2’s comments:

Reviewer #2: Referee report on Gender differences in dictator giving: a high-power laboratory test (Manuscript Number: PONE-D-24-05752)

This study investigates the presence of gender differences through lab experiments in the canonical game of altruism, the dictator game. As the authors note, the high number of observations allows for drawing firm conclusions in a topic on which there is not yet consensus. The authors document a significant gender difference, finding that females are more generous than males, even after accounting for a wide range of relevant control variables such as age, cognitive ability, emotions and personality traits.

This study is of high quality and makes a clear contribution to the literature, making it deserving of publication. However, there are some minor issues that should be addressed beforehand.

Authors: Many thanks for your positive appraisal of our work and for the suggestions. We have incorporated them into the revised version. Here are our responses. Your original comments are included for convenience, and our reply is in bold italics.

Comments

I did not understand how playing the dictator game in dual roles affects the results. The authors emphasize the relevance of dual roles, but based on the description of the experiment, I did not see why this is important. Perhaps this should be made clearer.

Authors: We have added a paragraph in the Experimental Design and Methods section mentioning the two possible differential effects of using a dual role version of the dictator game with respect

---

## [Decision Letter · Decision Letter 1]

18 Nov 2024

PONE-D-24-05752R1Gender differences in dictator giving: a high-power laboratory testPLOS ONE

Dear Dr. Pavan,

Thank you for submitting your manuscript to PLOS ONE. After careful consideration, we feel that it has merit but does not fully meet PLOS ONE’s publication criteria as it currently stands. Therefore, we invite you to submit a revised version of the manuscript that addresses the points raised during the review process.

Thank you for submitting your revised manuscript. While Reviewer #2 is satisfied with the changes, Reviewer #1, while appreciative of the improved introduction and attempted structural analysis, suggests further refinements to the SEM analysis - specifically recommending a simplified mediation analysis to better demonstrate how gender effects are mediated through personality traits, reasoning ability, and emotions. As editor, I fully concur with these suggestions and request that you address these final changes in your next revision, including adding the recommended footnote about the temporal relationship between emotions and decisions. Many thanks and alle the best, Holger Rau==============================

We look forward to receiving your revised manuscript.

Kind regards,

Holger A. Rau

Academic Editor

PLOS ONE

Journal Requirements:

Reviewers' comments:

Reviewer's Responses to Questions

**Comments to the Author**

1. If the authors have adequately addressed your comments raised in a previous round of review and you feel that this manuscript is now acceptable for publication, you may indicate that here to bypass the “Comments to the Author” section, enter your conflict of interest statement in the “Confidential to Editor” section, and submit your "Accept" recommendation.

Reviewer #1: (No Response)

Reviewer #2: All comments have been addressed

2. Is the manuscript technically sound, and do the data support the conclusions?

Reviewer #1: Yes

Reviewer #2: Yes

3. Has the statistical analysis been performed appropriately and rigorously? 

Reviewer #1: Yes

Reviewer #2: Yes

4. Have the authors made all data underlying the findings in their manuscript fully available?

Reviewer #1: Yes

Reviewer #2: Yes

5. Is the manuscript presented in an intelligible fashion and written in standard English?

Reviewer #1: Yes

Reviewer #2: Yes

6. Review Comments to the Author

Reviewer #1: The authors addressed most of my comments sufficiently. I think they have done a great job in making their contribution clearer in the introduction and I also, appreciate that they have tried to provide a bit more structure to their analysis by including a SEM. Still, this analysis should be explained better including the assumptions made. The way it is now, I fear that it makes the contribution of the paper harder to understand rather than improving clarity. I was asking for a much simpler mediation analysis showing how much of the effect of gender on the share sent is mediated through personality traits, reasoning ability and emotions, so more of a decomposition exercise (Gelbach, 2016) as for example done by Grosch and Rau (2016). This would simplify the analysis presented quite significantly.

I think it is possible to infer what I have asked for from the model the authors provide (and they state that it can be seen in Figure 5, but not where) but it just seems unnecessarily complicated.

Also, the authors should at least acknowledge in a footnote that they have no way of knowing whether the emotions subjects state to have had were causing the decision or whether their reporting ex-post is affected by the decision made. For example, subjects who decided to give nothing might feel better to state that this was rational no matter what was actually driving their decision.

References:

Kerstin Grosch, Holger A. Rau (2017). Gender differences in honesty: The role of social value orientation. Journal of Economic Psychology 62, 258-267

2017, Pages 258-267Gelbach, J. B. (2016). When do covariates matter? And which ones, and how much? Journal of Labor Economics 34 (2), 509–543.

Reviewer #2: My concerns have been properly addressed. This paper will contribute to our understanding of gender differences in dictator games.

7. PLOS authors have the option to publish the peer review history of their article (what does this mean?). If published, this will include your full peer review and any attached files.

Reviewer #1: No

Reviewer #2: **Yes: **Hubert János Kiss

---

## [Author Response · Author response to Decision Letter 1]

21 Dec 2024

Editor:

Dear Dr. Pavan,

Thank you for submitting your manuscript to PLOS ONE. After careful consideration, we feel that it has merit but does not fully meet PLOS ONE’s publication criteria as it currently stands. Therefore, we invite you to submit a revised version of the manuscript that addresses the points raised during the review process.

Thank you for submitting your revised manuscript. While Reviewer #2 is satisfied with the changes, Reviewer #1, while appreciative of the improved introduction and attempted structural analysis, suggests further refinements to the SEM analysis - specifically recommending a simplified mediation analysis to better demonstrate how gender effects are mediated through personality traits, reasoning ability, and emotions. As editor, I fully concur with these suggestions and request that you address these final changes in your next revision, including adding the recommended footnote about the temporal relationship between emotions and decisions.

Many thanks and alle the best, 

Holger Rau

Authors: Dear Editor, many thanks for giving us a second opportunity to improve our work. In this version, we simplified the SEM analysis and added a limitation of our measure of emotions. We answer to the reviewer’s comments below.

Reviewer #1: The authors addressed most of my comments sufficiently. I think they have done a great job in making their contribution clearer in the introduction and I also, appreciate that they have tried to provide a bit more structure to their analysis by including a SEM. Still, this analysis should be explained better including the assumptions made. The way it is now, I fear that it makes the contribution of the paper harder to understand rather than improving clarity. I was asking for a much simpler mediation analysis showing how much of the effect of gender on the share sent is mediated through personality traits, reasoning ability and emotions, so more of a decomposition exercise (Gelbach, 2016) as for example done by Grosch and Rau (2016). This would simplify the analysis presented quite significantly.

I think it is possible to infer what I have asked for from the model the authors provide (and they state that it can be seen in Figure 5, but not where) but it just seems unnecessarily complicated.

Authors: Dear Reviewer, we thank you for your constructive comments and hope to have improved our work. In particular, we have simplified the SEM analysis as you suggested. We now present a simpler SEM model to test whether the gender effect is mediated by personality traits, reasoning ability and emotions. Results do not differ from our previous analysis, but we hope that they are more easily interpretable now. 

Reviewer #1: Also, the authors should at least acknowledge in a footnote that they have no way of knowing whether the emotions subjects state to have had were causing the decision or whether their reporting ex-post is affected by the decision made. For example, subjects who decided to give nothing might feel better to state that this was rational no matter what was actually driving their decision.

Authors: Thank you for the suggestion. Since it is not possible to include footnotes in PLOS-ONE articles, we added a comment at page 18 of the paper, when describing our measure of emotions:

“A possible limitation of this measure is that we cannot be sure whether the emotions subjects stated to have had were causing the decision or whether their reporting ex-post was affected by the decision made.”

---

## [Decision Letter · Decision Letter 2]

8 Jan 2025

Gender differences in dictator giving: a high-power laboratory test

PONE-D-24-05752R2

Dear Dr. Pavan,

We’re pleased to inform you that your manuscript has been judged scientifically suitable for publication and will be formally accepted for publication once it meets all outstanding technical requirements.

Kind regards,

Holger A. Rau

Academic Editor

PLOS ONE

Additional Editor Comments (optional):

Reviewers' comments:

Reviewer's Responses to Questions

**Comments to the Author**

1. If the authors have adequately addressed your comments raised in a previous round of review and you feel that this manuscript is now acceptable for publication, you may indicate that here to bypass the “Comments to the Author” section, enter your conflict of interest statement in the “Confidential to Editor” section, and submit your "Accept" recommendation.

Reviewer #1: All comments have been addressed

2. Is the manuscript technically sound, and do the data support the conclusions?

Reviewer #1: Yes

3. Has the statistical analysis been performed appropriately and rigorously? 

Reviewer #1: Yes

4. Have the authors made all data underlying the findings in their manuscript fully available?

Reviewer #1: Yes

5. Is the manuscript presented in an intelligible fashion and written in standard English?

Reviewer #1: Yes

6. Review Comments to the Author

Reviewer #1: Thank you for addressing my concerns in such a constructive way. Congratulations to a nice publication!

7. PLOS authors have the option to publish the peer review history of their article (what does this mean?). If published, this will include your full peer review and any attached files.

Reviewer #1: No

---

## [Editor Report · Acceptance letter]

13 Jan 2025

PONE-D-24-05752R2 

PLOS ONE

Dear Dr. Pavan, 

I'm pleased to inform you that your manuscript has been deemed suitable for publication in PLOS ONE. Congratulations! Your manuscript is now being handed over to our production team.

Kind regards, 

on behalf of

Prof. Dr. Holger A. Rau 

Academic Editor

PLOS ONE